# Rapid increase in dichloromethane emissions from China inferred through atmospheric observations

Minde An [1,2], Luke M. Western [2], Daniel Say [2], Liqu Chen[3], Tom Claxton [4], Anita L. Ganesan [5], Ryan Hossaini [4,6], Paul B. Krummel [7], Alistair J. Manning [8], Jens Mühle [9], Simon O'Doherty [2], Ronald G. Prinn [10], Ray F. Weiss [9], Dickon Young [2], Jianxin Hu [1✉], Bo Yao [3,11✉] & Matthew Rigby [2✉]

With the successful implementation of the Montreal Protocol on Substances that Deplete the Ozone Layer, the atmospheric abundance of ozone-depleting substances continues to decrease slowly and the Antarctic ozone hole is showing signs of recovery. However, growing emissions of unregulated short-lived anthropogenic chlorocarbons are offsetting some of these gains. Here, we report an increase in emissions from China of the industrially produced chlorocarbon, dichloromethane ($CH_2Cl_2$). The emissions grew from 231 (213–245) Gg $yr^{-1}$ in 2011 to 628 (599–658) Gg $yr^{-1}$ in 2019, with an average annual increase of 13 (12–15) %, primarily from eastern China. The overall increase in $CH_2Cl_2$ emissions from China has the same magnitude as the global emission rise of 354 (281—427) Gg $yr^{-1}$ over the same period. If global $CH_2Cl_2$ emissions remain at 2019 levels, they could lead to a delay in Antarctic ozone recovery of around 5 years compared to a scenario with no $CH_2Cl_2$ emissions.

[1] College of Environmental Sciences and Engineering, Peking University, Beijing, China. [2] School of Chemistry, University of Bristol, Bristol, UK. [3] Meteorological Observation Centre of China Meteorological Administration (MOC/CMA), Beijing, China. [4] Lancaster Environment Centre, Lancaster University, Lancaster, UK. [5] School of Geographical Sciences, University of Bristol, Bristol, UK. [6] Centre of Excellence in Environmental Data Science, Lancaster University, Lancaster, UK. [7] Climate Science Centre, CSIRO Oceans and Atmosphere, Aspendale, VIC, Australia. [8] Hadley Centre, Met Office, Exeter, UK. [9] Scripps Institution of Oceanography, University of California San Diego, La Jolla, CA, USA. [10] Center for Global Change Science, Massachusetts Institute of Technology, Cambridge, MA, USA. [11] Department of Atmospheric and Oceanic Sciences & Institute of Atmospheric Sciences, Fudan University, Shanghai, China. ✉email: jianxin@pku.edu.cn; yaob@cma.gov.cn; matt.rigby@bristol.ac.uk

Global emissions of long-lived ozone-depleting substances (ODSs) such as chlorofluorocarbons (CFCs), halons, hydrochlorofluorocarbons (HCFCs) and carbon tetrachloride (CCl₄), which are regarded as the main contributors to stratospheric ozone depletion, have decreased significantly as a result of regulations imposed by the Montreal Protocol and its amendments[1]. This has led to reductions in stratospheric bromine and chlorine abundances and the onset of recovery of the Antarctic ozone hole[2,3]. The remaining uncertainties concerning global ozone layer recovery partly originate from very short-lived halogenated substances (VSLS), defined as species with an atmospheric lifetime shorter than ~6 months[1]. Previously, VSLS were thought to have a minor influence on stratospheric chlorine and bromine levels and hence are not regulated under the Montreal Protocol. However, recent studies have found substantial and growing contributions of VSLS[4–10] to stratospheric ozone depletion, which could offset some of the benefits of the Montreal Protocol, particularly when emissions are from regions such as East and South Asia, where strong convective systems facilitate their rapid transport into the stratosphere[4,8,11–15].

Dichloromethane (CH₂Cl₂), the most abundant chlorine-containing VSLS with a lifetime of ~6 months[16], accounts for ~70% of the total stratospheric source gas injection from chlorine-containing VSLS[1,7]. This substance originates mainly from anthropogenic sources, including its use as an emissive solvent for adhesive and cleaning purposes, and as a feedstock for hydrofluorocarbon (HFC) production[17–19]. Measurements of the atmospheric mole fraction of CH₂Cl₂ show a rapid rise since the 2000s, where the annual global mean values have undergone a twofold increase, including a period of particularly rapid growth during 2012–2013[1,20]. A global chemical transport model sensitivity study[6] estimated a substantial delay in the recovery of the Antarctic ozone layer, by up to ~30 years, if CH₂Cl₂ mole fraction growth continued at the rate observed between 2004 and 2014. The significant increase in global emissions of CH₂Cl₂, from 637 (600−673) Gg yr$^{-1}$ (1 s.d. uncertainty) in 2006 to 1171 (1126−1216) Gg yr$^{-1}$ in 2017, was attributed to an increase in industrial emissions from Asia[21]. As emissions from East and South Asia can be rapidly transported to the stratosphere by convective systems, it is critical to quantify emissions from this region to help understand its growing impact on stratospheric ozone. However, there are few atmospheric observation-derived (top-down) estimates of CH₂Cl₂ emissions within Asia, and no estimates that span multiple years.

In this study, we infer a substantial increase in the annual CH₂Cl₂ emissions from China (defined as the Chinese mainland, excluding Hong Kong and Macao) in 2011–2019, using measurements from nine sites within the country and an inverse modelling approach. This top-down time series agrees well with a bottom-up inventory compiled using newly available consumption and production data. We find that the increase in emissions from China plays an important role in the global emissions growth, and these increases have the potential to impact the recovery of the stratospheric ozone layer.

## Results

### Global emissions of CH₂Cl₂.
Hemispheric CH₂Cl₂ mole fractions (Fig. 1) were estimated by assimilating baseline atmospheric measurements from the Advanced Global Atmospheric Gases Experiment (AGAGE)[22] into a 12-box model of atmospheric transport and chemistry[23,24] (see Methods). Annual averages of observed mole fractions rose continuously between 2011–2019 (inclusive), with a larger growth observed in the Northern Hemisphere than in the Southern Hemisphere. During this period, average annual growth rates were estimated to be

2.29 (2.01–2.58) and 0.71 (0.65–0.78) ppt yr$^{-1}$ (68% uncertainty) for the Northern and Southern Hemispheres, respectively. The highest growth rate in both hemispheres occurred at 2012–2013, with a maximum global average growth rate of 4.43 (4.07–4.81) ppt yr$^{-1}$. The large and increasing inter-hemispheric gradient for CH₂Cl₂ indicates ongoing growth in Northern Hemispheric emissions, relative to those in the Southern Hemisphere. Global emissions derived from the 12-box model and AGAGE data (Fig. 2a) are an update to previously published emissions through 2016[1]. Emissions have grown substantially, from 683 (541–825) Gg yr$^{-1}$ (68% uncertainty) in 2011 to 1038 (826–1251) Gg yr$^{-1}$ in 2019, sustaining the trend observed before 2016, albeit with a lower emission growth rate after 2017. This estimate shows a similar increase to recently published global emissions, derived using TOMCAT (a global 3D model) with measurement data from both the US National Oceanic and Atmospheric Administration (NOAA) and AGAGE (adjusted to the NOAA calibration scale)[21], or derived using a 12-box model with measurement data from NOAA alone[1]. The different global emissions estimates agree within 1 s.d. uncertainty range, although the means differ by ~10–20%, partly due to the differences in calibration scales between NOAA and AGAGE (~10% in CH₂Cl₂ measurement) and differences in the locations of the measurement sites used in the inversion[1]. The majority of the global growth has previously been attributed to increasing industrial emissions from Asia[21]. As China has been shown to be a major contributor to halocarbon emissions in Asia, and with its CH₂Cl₂ emissions projected to increase in the future[17,25], we focus on emissions from China in this study.

### Emissions of CH₂Cl₂ from China.
Emissions of CH₂Cl₂ from China were derived from atmospheric mole fractions observed at nine remote sites within the China Meteorological Administration's (CMA) monitoring network by an inverse modelling approach (see Methods for site information, and see Supplementary Fig. 1 and Supplementary Data 1 for mole fractions). The inferred CH₂Cl₂ emissions from China have increased from 231 (213–245) Gg yr$^{-1}$ in 2011 to 628 (599–658) Gg yr$^{-1}$ in 2019, with an overall increase of 173 (149–195) % (Fig. 2a). The mean annual emissions growth rate is 13 (12–15) %. There was a rapid increase in emissions after 2012, from 272 (247–291) Gg yr$^{-1}$ in 2012 to 534 (477–574) Gg yr$^{-1}$ in 2015, coinciding with the largest observed global CH₂Cl₂ mole fraction growth rate. After 2015, emissions continued to rise overall, but at a much slower rate. Our modelled emissions are found to be relatively insensitive to the a priori emissions estimate and its uncertainty used in the inversion framework (Supplementary Fig. 2), and to the increasing number of measurement sites in the inversion throughout the study period (Supplementary Fig. 3).

China is a major contributor to global halocarbon emissions[25]. By comparing our regional and global estimates (Fig. 2a), we find that China accounted for ~30–35% of global CH₂Cl₂ emissions in 2011–2012. After 2012, emissions from China accounted for ~50–60% of the global total. The inferred emissions of CH₂Cl₂ from China and the inferred global emissions in this study follow similar trends, and the overall increase in CH₂Cl₂ emissions from China during the inversion period, 397 (363–430) Gg yr$^{-1}$ between 2011–2019, has the same magnitude as the total global increase, 354 (281−427) Gg yr$^{-1}$. These results strongly suggest that China is the dominant source for the global emissions increase over this study period.

There is only one existing time series for CH₂Cl₂ emissions from China derived in Feng et al.[17], which is an inventory-based study. Between 2011 and 2012, our top-down estimates (inversion) are similar to that bottom-up estimate (inventory) (Fig. 2b).

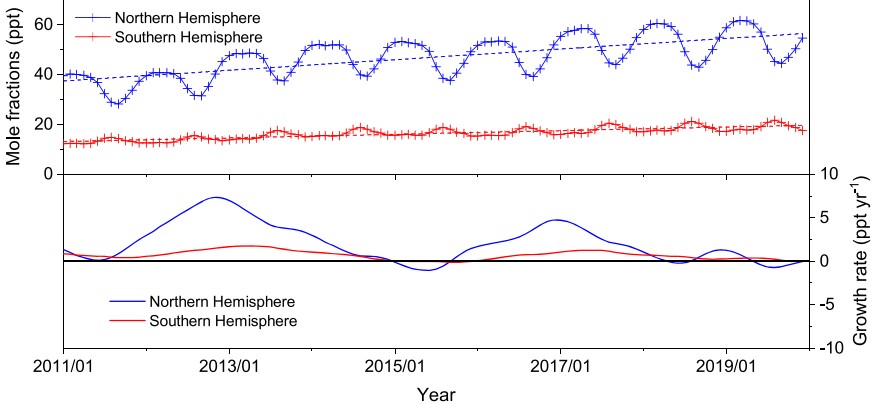

**Fig. 1 Hemispheric mole fractions of CH₂Cl₂ and their growth rate.** Global mole fractions of $CH_2Cl_2$ (2011–2019) were inferred using the AGAGE 12-box model and data from 5 AGAGE background sites (see Methods). The upper panel shows the mole fractions in each hemisphere and their trends (dashed lines). The lower panel shows the growth rate in each hemisphere, with a smoothing timescale of ~1.4 years[67].

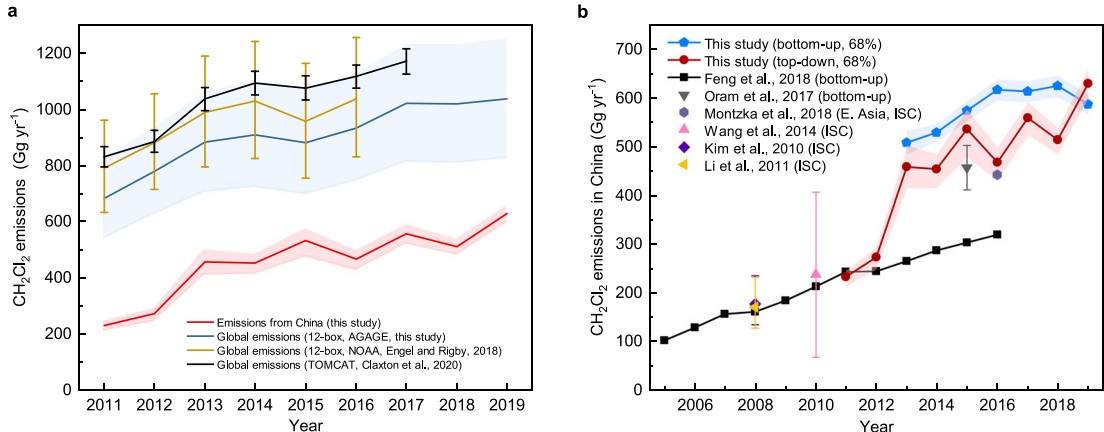

**Fig. 2 Global emissions and emissions from China. a** Derived global emissions of $CH_2Cl_2$ in 2011–2019 and uncertainties (blue line and shading), and emissions from China (top-down, inversion based) and uncertainties (red line and shading). Global emissions derived using the 12-box model with NOAA data[1] (yellow line), and by TOMCAT (a global 3D model) with data from multiple sources[21] (black line), are shown in the plot for comparison. All uncertainties are the 68% interval. **b** Comparison of top-down emissions from China derived in this study (red line and shading) to previous bottom-up (inventory based) time series[17] (time period overlap 2011–2016, black line), and other estimates for specific years[4,29,68–70]. The result from Montzka et al.[29] is the regional $CH_2Cl_2$ emissions for East Asia. ISC means "interspecies correlation" method. A new bottom-up analysis for 2013–2019 (blue line and shading in **b**) is estimated using newly obtained consumption and production data from CCAIA[33] (data shown in Supplementary Fig. 4a). Bottom-up results for individual sectors are shown in Supplementary Fig. 4b. All emissions results estimated in this study can be found in Supplementary Table 5.

However, after 2012, our top-down estimates increased much more rapidly, leading to a large discrepancy during later years. Similar discrepancies between top-down and bottom-up estimates have been observed for other substances, such as CCl₄[26], trichlorofluoromethane (CFC-11)[25,27–30], and some HFCs[31,32], and may partially be explained by unknown sources, or inaccurate activity data or emission factors. Our results agree well with the bottom-up emissions estimate for 2015 by Oram et al.[4], who used the reported production of HCFC-22 (CHClF₂) in China to deduce the chloroform (CHCl₃) production needed for this amount of HCFC-22, and then used the production ratio of CH₂Cl₂ to CHCl₃ to estimate CH₂Cl₂ emissions in China. The derived production data of CH₂Cl₂ by Oram et al.[4] are much higher than the production values used in Feng et al.[17]. Another independent estimate[29] based on mole fraction ratios of CH₂Cl₂/HCFC-22 measured at the NOAA Mauna Loa Observatory in Hawaii gives a regional emission of CH₂Cl₂ from East Asia of 440 Gg yr⁻¹ in 2016, which is also consistent with our results.

To reconcile the discrepancy with the bottom-up emission time series by Feng et al.[17], a new bottom-up emission inventory was estimated following the method in Feng et al.[17] (with some modifications, see Methods), using newly obtained production and consumption data for 2013–2019 from the China Chlor-Alkali Industry Association (CCAIA)[33]. CCAIA is the only alkali industry association in China, and all chloromethanes companies/manufacturers are its members. These companies have the responsibility to share their production data with the association, ensuring that the dataset is representative of China's entire chloro-alkali industry. These newly available production and consumption estimates are approximately twice as large as those used by Feng et al.[17] (Supplementary Fig. 4a). Therefore, our new bottom-up emissions estimates are significantly larger (results shown in Fig. 2b, sectoral results in Supplementary Fig. 4b). The discrepancies between the two bottom-up inventories mainly originate from the emissive solvent sector, due to the difference in production and consumption data used in the two studies. According to the bottom-up inventory, the solvent sector accounts for more than 90% of overall CH₂Cl₂ emissions from China in all years and more than 85% of the overall increase during the study period, which indicates that the solvent sector is

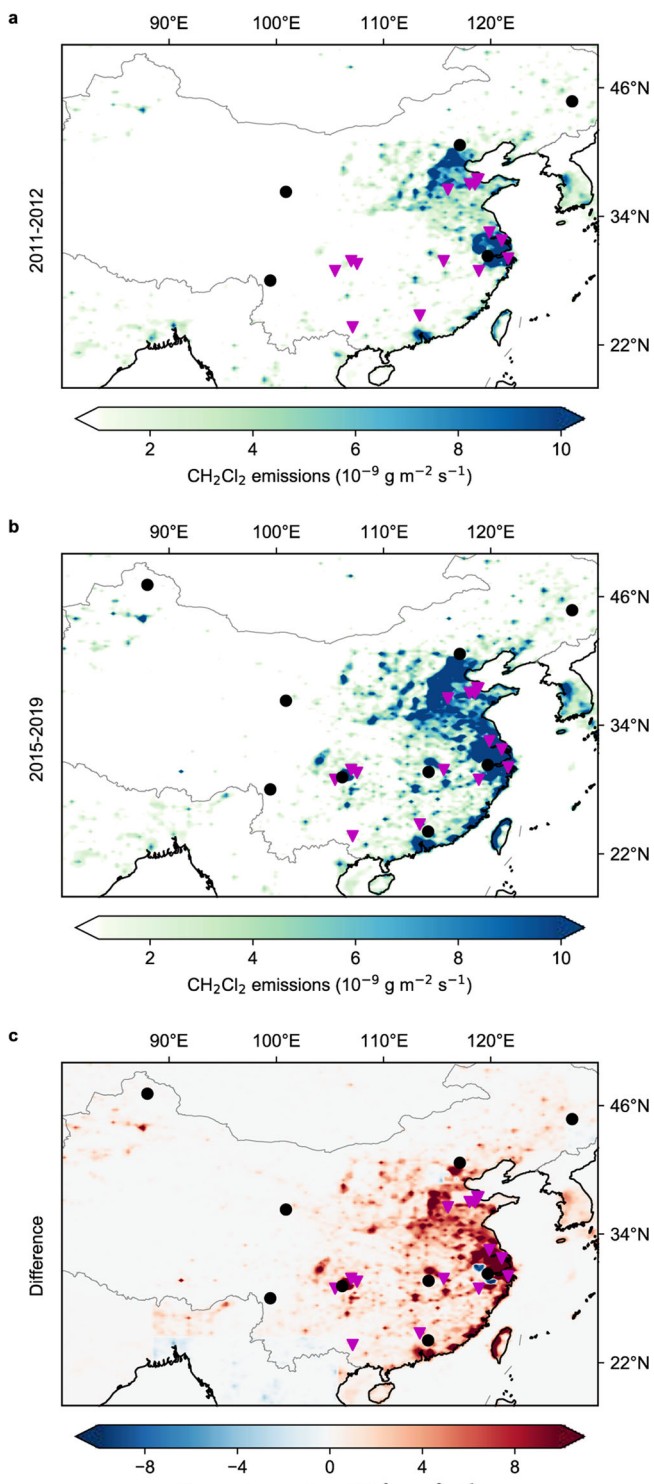

**Fig. 3 Spatial distribution of CH$_2$Cl$_2$ emissions in Eastern Asia with focus on China. a** The average mean emissions of CH$_2$Cl$_2$ in 2011–2012. **b** The average mean emissions of CH$_2$Cl$_2$ in 2015–2019. **c** The difference between **a** and **b**. The two time periods are divided by the rapid increase in emissions from China which occurred in 2012–2015. Black dots in the figures represent measurement sites active during that time period; pink triangles are the known chloromethanes factories in China. The spatial distribution for each year is shown in Supplementary Fig. 8. The difference between the spatial distribution of the mean top-down and a priori emission is shown in Supplementary Fig. 9.

the dominant source for the emissions of CH$_2$Cl$_2$ from China. Our new bottom-up emissions are ~10–20% higher than our top-down estimates, which may be due to the assumption made in our bottom-up estimate that all consumption, except feedstock use for HFC-32 (CH$_2$F$_2$) and use in pharmaceutical production, was regarded as 100% emissive. Some CH$_2$Cl$_2$ may be used in the synthesis of fine chemicals, despite being affiliated with the emissive solvent sector in our analysis, which would result in a lower overall emissivity. Furthermore, the CH$_2$Cl$_2$ used as blowing agent for polyurethane (PU) foams, which was included in the emissive solvent use sector in this study, may not always result in a complete release at the time of use. Our bottom-up estimates should therefore be considered as an upper limit on China's CH$_2$Cl$_2$ emissions. Additionally, there may be a lag between production and consumption, leading to a delay in emissions, which we are unable to account for here.

**Spatial distribution of emissions in China**. The eastern part of China, including part of the North China Plain and the Yangtze River Delta region, are shown to be the main source regions for CH$_2$Cl$_2$ over the study period (Fig. 3). These regions also contribute most to the increase between the pre-2012 and post-2015 periods (Fig. 3c), especially the Yangtze River Delta region, which consists of the highly populated provinces, Zhejiang, Jiangsu, Shanghai and Anhui. Annual provincial emissions are presented in the Supplementary Information (Supplementary Data 2). Emissions from Jiangsu and Zhejiang are among the highest over the study period, together contributing ~20–30% of the national total emissions, and also have the largest increase between the two periods, at 37 (30–44) Gg yr$^{-1}$ and 24 (18–30) Gg yr$^{-1}$, or rate of 164 (104–200) % and 85 (53–109) %, respectively. As 70–90% of global CH$_2$Cl$_2$ emissions are estimated to be from anthropogenic sources[19,34], the finding of high emissions from these major population centers is not surprising. Likewise, there are major emissions from Shandong and Hebei, highly industrialized regions located in the North China Plain, where high levels of halocarbons, including CH$_2$Cl$_2$, have previously been detected in the atmosphere[35]. Shandong and Hebei together contribute ~15–20% of the national total emissions, with increase between the two periods of 20 (14–25) Gg yr$^{-1}$ and 18 (14–21) Gg yr$^{-1}$, or rate of 82 (46–107) % and 65 (47–80) %, respectively. There is an increase in emissions from the Sichuan Basin (located approximately at 103–108° E, 28–32° N) between the two periods (Fig. 3c), although this is relatively uncertain due to the lack of measurements from the nearby Jiangjin (JGJ) site before 2017. The spatial distribution of regions with high emissions or large growth over the period are insensitive to the a priori distribution of emissions used in the inversion (Supplementary Fig. 5). Most chloromethanes factories in China are in Shandong province, the Yangtze River Delta and the Sichuan Basin (Fig. 3), which is consistent with the key regions that exhibit high emissions and emission increase in this study. Previous studies[36,37] have reported substantial fugitive emissions from chloromethanes plants.

## Discussion

There is strong evidence that Cl-containing VSLS, especially CH$_2$Cl$_2$, contribute significantly to stratospheric chlorine[6,7] and thereby stratospheric ozone depletion[6,8,20]. The contribution of Cl-VSLS to total stratospheric chlorine has increased by up to 50% in recent decades[1,7]. In this study, a substantial increase in global CH$_2$Cl$_2$ mole fractions and emissions was observed over 2011–2019, which is currently dominated by the growth in emissions from China. The average global mole fraction growth

rate between 2011 and 2019 is close to a "high growth" scenario used in a model study by Hossaini et al.[6], which resulted in a delay of 17–30 year in Antarctic ozone recovery, assuming mole fractions (or emissions) continued to rise at this rate (see the first scenario in Methods). As we show here, a slowdown in the increase of both global emissions and emissions from China seems to have occurred during the later years of our study period, which may indicate that such extreme growth rates will not be sustained. If instead of growing further, global emissions remain close to 2019 levels in the future, the recovery of Antarctic stratospheric ozone back to 1980 levels could be delayed by ~5 years compared to the scenario with no $CH_2Cl_2$ emissions from Hossaini et al.[6] (see the second scenario in Methods). These impacts are comparable to or even greater than the delay in Antarctic ozone recovery caused by the recent unexpected CFC-11 emissions increase[38] or the recently identified rise in $CHCl_3$ emissions[39]. It is worth emphasizing that our estimates are based on results from a sensitivity study[6], where the delay of Antarctic ozone recovery was calculated in reference to a scenario with no $CH_2Cl_2$ emissions. A more precise quantification of the impacts of only the emissions from China, or the impact of the emissions rise between 2011 and 2019, would require a dedicated model study, which is beyond the scope of this work.

The main regions of $CH_2Cl_2$ emissions in China and their increase over the study period are highly consistent with the locations of economically developed and industrialized regions, confirming a substantial anthropogenic source for emissions in China. This finding is supported by the bottom-up inventory in this study, where emissive solvent use of $CH_2Cl_2$ such as painting or adhesive use, accounts for most of the emissions and their increase. The main regions are also consistent with locations of known chloromethanes factories (shown in Fig. 3). Notably, substantial emissions of two other chloromethanes, $CHCl_3$[39] and $CCl_4$[27,40], were recently reported from these same regions. The timing of increased emissions of $CHCl_3$ and $CCl_4$ is also similar to the increase of $CH_2Cl_2$ found in this study, though emissions of $CCl_4$ declined in 2017, unlike emissions of $CH_2Cl_2$. These correlations suggest a common link between emissions of $CH_2Cl_2$ and the wider chloromethanes industry in China. Since $CH_2Cl_2$, $CHCl_3$, and $CCl_4$ are produced simultaneously at a ratio of between 40:60 to 60:40 for $CH_2Cl_2/CHCl_3$ (with $CCl_4$ produced as an unavoidable byproduct at ~4%[41]), an increase in production of one inevitably leads to an increase in production for all chloromethanes. The increase in general chloromethanes production, which could be driven by the growing economy in China, would therefore be expected to cause an increase in $CH_2Cl_2$ emissions. This explains the strong correlation between chloromethanes emissions, Gross Domestic Product and the expanding chloromethanes production in China (Supplementary Fig. 6).

There are indications that emissions may not continue to grow as rapidly in the near future. The total chloromethanes production in China decreased between 2018 and 2019 due to current oversupply and low profit margins within the industry[33]. Emissions of $CH_2Cl_2$ from China are dominated by emissive solvent use and the PU foam sector, followed by pharmaceutical use, production leakage and feedstock use (Supplementary Fig. 4b, also see Feng et al.[17]). The use of $CH_2Cl_2$ is limited in several sectors by recently published national regulations in China, such as in the pharmaceutical[42], painting[43] and adhesive[44] industries, as part of volatile organic compounds (VOCs) control measures, which are similar to regulations in the U.S[45]. and Europe[46]. However, the current regulations in China only pose restrictions on concentrations of $CH_2Cl_2$ in consumer products or release rates from industrial processes, and no limits on overall production or consumption. There are emerging replacements for $CH_2Cl_2$ in many sectors in order to comply with these regulations, in large parts to avoid the toxicity of $CH_2Cl_2$, including esters in adhesives, and methylbenzene (toluene) diluents and water-based cleaners in emissive solvent uses[33]. Hence future demand for $CH_2Cl_2$ in these sectors is likely to decline. In contrast, feedstock use of $CH_2Cl_2$ for HFC-32 production is likely to increase in the coming years, driven by the increasing demand for low-GWP, relatively short-lived and non-ozone-depleting refrigerants. Strong correlation between enhanced levels of HFC-32 and $CH_2Cl_2$ in India[18] was seen as an indication of $CH_2Cl_2$ emissions from HFC-32 production related activities.

With emissions of $CH_2Cl_2$ in Europe and North America declining[21], emissions from the developing world will have a growing impact on global $CH_2Cl_2$ emissions. In addition to the growth in emissions from China, a potential increase in $CH_2Cl_2$ emissions from India has been identified, based on emission estimates of 20.3 Gg yr$^{-1}$ in 2008[47] and 96.5 Gg yr$^{-1}$ in 2016[18]. The magnitude of this rise is relatively uncertain due to the methodological differences in the two studies, and it is small compared to the inferred increase from China. However, given that our study indicates that the growth in emissions from China is consistent with the coincident global rise, it is possible that any growth in India's emissions has offset a decline from North America and Europe. Using our estimated emissions for China and the estimates for India in 2016[18], they together accounted for ~60% of the total Asian emission estimated by Claxton et al.[21]. The remaining ~40% likely originates from a combination of emissions from both land and ocean, although the exact emission breakdown is very uncertain due to the differences in methodology and measurement calibration used in the different studies. Emissions from East and South Asia have the potential to enter the stratosphere more quickly than emissions from other parts of the world due to the Asian monsoon circulation[4,8,11–15], and therefore pose a greater threat to stratospheric ozone than similar emissions from other regions. Given that atmospheric mole fractions and emissions of many long-lived ozone-depleting substances have declined substantially as a result of the Montreal Protocol, the impact of unregulated $CH_2Cl_2$ on the ozone layer, which in this study is estimated to delay Antarctic ozone recovery by 5–30 years depending on different future scenarios, is of increasing importance. Should emissions continue to grow, $CH_2Cl_2$ could rival that of controlled ODSs (e.g. CFCs and HCFCs) in coming decades[6,36]. Thus, continued or expanded monitoring of $CH_2Cl_2$ and other VSLS, especially in East and South Asia, will be required to determine their evolving contribution to global ozone depletion.

## Methods

**Sampling and analysis.** Atmospheric mole fraction observations were conducted at nine stations located around China, which are operated by the China Meteorological Administration (CMA). The sites include Akedala (AKD) in Xinjiang province, Northwest China, Lin'an (LAN) in the Yangtze River Delta region, East China, Jiangjin (JGJ) in Sichuan Basin, Southwest China, Shangri-La (XGL) on the Yunnan-Guizhou Plateau, Southwest China, Jinsha (JSA) in Central China, Longfengshan (LFS) on the Northeast China Plain, Mt. Waliguan (WLG) on the Qinghai-Tibet Plateau, Northwest China, Xinfeng (XFG) in the Pearl River Delta region, South China and Shangdianzi (SDZ) on the North China Plain. The station information is summarized in Supplementary Table 1. Detailed descriptions for LAN, LFS, WLG, SDZ, XGL and JGJ can be found in previous studies[48,49]. All sites except JGJ and LAN are more than 20 km away from the nearest industrial areas and situated in elevated positions in order to sample background, well-mixed air, while JGJ and LAN are located ~10 km from their nearest industrial area.

The sampling and analysis procedure has been previously described[48,49], and is briefly summarized here. Weekly flask air samples were collected at AKD, LFS, JSA, SDZ, WLG, XFG and XGL, and daily samples at JGJ. For LAN, air samples were taken weekly before December 2018 and daily thereafter. Ambient air was pumped through a 10 mm OD sampling tube (Synflex-1300, Eaton, USA) into 3-L stainless steel canisters (X23-2N, LabCommerce, Inc., USA) from the tops of the towers at each sampling site using a membrane pump (KNF-86, KNF Neuberger, Germany), and then sent to the CMA lab in Beijing. The dry-air mole fractions of a wide suite

of trace gases, including $CH_2Cl_2$, were then measured in each air sample. In addition to the above station sampling, in situ measurements were made every 2 h at SDZ before August of 2012 and after December of 2015, as part of the AGAGE network[22]. The sampling procedure for in situ measurements is the same as for the flask air sampling mentioned above, except that air was pumped from a nearby tower into the analysis system directly. The 3-year gap in SDZ in situ data was due to system malfunction. All flask air and in situ analyses were conducted using a 'Medusa' gas chromatographic system with mass spectrometric detector (Agilent 6890/5975B, USA)[50,51]. Each 2 L ambient air sample was bracketed by analyses of a reference gas (the working standard) to account for short-term instrumental drift. All $CH_2Cl_2$ measurements were reported relative to the SIO-14 (Scripps Institution of Oceanography) calibration scale[22]. The repeatability for $CH_2Cl_2$ measurements is estimated to be 0.8 and 2% for in situ and flask air measurements, respectively. The in situ measurements from SDZ were averaged every 24 h to reduce the correlation within the in situ measurements and the computational cost in the inversion. As a result, a total of 4661 measurements were used in the inversion, after resampling.

The measurements made at these sites are sensitive to surface emissions from most of China (Supplementary Fig. 7), which makes it feasible to derive total national emissions from China using inverse modelling. The mean sensitivity from the sites in each year did not change substantially throughout the period, even though several new measurement sites were established and incorporated after 2017. The regional inversion was repeated using only observations from the five sites that were operational throughout the study period to show that the derived regional emissions are relatively insensitive to the number of measurement sites in the inversion (Supplementary Fig. 3).

**Regional transport models and sensitivity.** Exploiting the linear relationship between emissions and observed mole fractions, the forward model can be expressed as:

$$\mathbf{y} = \mathbf{Hx} + \mathbf{e} \qquad (1)$$

where $\mathbf{y}$ is a vector containing the observations; $\mathbf{x}$ describes a scaling of an a priori estimate; $\mathbf{H}$ is the corresponding sensitivity matrix, representing the sensitivities of atmospheric observations to the surface emissions within the regional domain and to the boundary conditions at the domain edge at that observation time; $\mathbf{e}$ represents the random error component.

In this study, sensitivities were estimated by 30-day backward trajectories output from the UK Met Office Numerical Atmospheric-dispersion Modelling Environment (NAME)[52] model. NAME is a Lagrangian Particle Dispersion Model that simulates the advection and turbulent diffusion of hypothetical particles in the atmosphere. The meteorological fields used to drive the model were obtained from the UK Met office Unified Model[53], with increasing spatial resolution throughout the period of study from 0.352° to 0.141° longitude and 0.234° to 0.094° latitude, and a constant temporal resolution of 3 h. The computational domain in the study was chosen to be bounded at 5° S and 74° N and 55° E and 192° E, which is sufficiently large to simulate particle transport in China. At each site, particles were released at a rate of 20,000 particles hour$^{-1}$ from the sampling inlet within a ± 10 m vertical range and then transported backward in time for 30 days (or, until the particles left the domain, which was the case for the vast majority of particles). All particles were considered inert over the duration of a given simulation. While the lifetime of $CH_2Cl_2$ does permit for some chemical loss during a 30-day simulation, previous work[39] showed that inclusion of a chemical loss scheme for gases with lifetimes of ~5–6 months did not significantly alter the inverse results (<1%, which is substantially smaller than other estimated uncertainties in the system). Particles were assumed to interact with surface emissions when they were in the lowest 40 m of the atmosphere[54].

**Regional a priori fluxes and boundary conditions.** The initial estimates of total a priori emissions for China and for the whole domain were adapted from a bottom-up inventory analysis of $CH_2Cl_2$ emissions in China[17], and a top-down estimate of global $CH_2Cl_2$ emissions[21] (which did not use any of the data from our study), respectively. The values are listed in Supplementary Table 2. The values inside China and outside China (defined as the values of the whole domain minus values of China) were distributed across the underlying grid inside and outside China independently of each other as a function of nightlight density data taken from the NOAA DMSP-OLS (Defense Meteorological Program-Operational Line-Scan System, https://ngdc.noaa.gov/eog/data/web_data/v4composites/). Nightlights (i.e. anthropogenic lighting observable from space during darkness) constitute an approximate representation of population and industrialization density and are therefore assumed to be a reasonable proxy for $CH_2Cl_2$ emissions[55]. The underlying grid was aggregated into 150 basis functions using a quadtree algorithm[56], determined by the a priori contribution of each region to the mole fraction enhancement. This algorithm results in basis functions of higher resolution in regions with larger a priori contribution to the above baseline mole fraction (i.e. those that are close to the measurement sites, and/or those that have high emissions). The monthly mole fraction values on the four boundaries of the domain were estimated by the TOMCAT atmospheric transport model[57]. The a priori mole fractions at the domain edge were interpolated onto the NAME output resolution using their nearest neighbor.

**Regional inversion theoretical framework.** A hierarchical Bayesian inference methodology was utilized in this study to estimate $CH_2Cl_2$ emissions, as shown by Eq. (2).

$$p(\mathbf{x}, \boldsymbol{\theta}|\mathbf{y}) \propto p(\mathbf{y}|\mathbf{x}, \boldsymbol{\theta})p(\mathbf{x}, \boldsymbol{\theta}) \qquad (2)$$

In the equation, $p(\mathbf{x}, \boldsymbol{\theta}|\mathbf{y})$ is the posteriori probability of $\mathbf{x}$, which contains emissions and boundary conditions, and the hyperparameters, $\boldsymbol{\theta}$, represent the uncertain model error. Measurement data are stored in the vector, $\mathbf{y}$. $p(\mathbf{y}|\mathbf{x}, \boldsymbol{\theta})$ is the likelihood, which follows a multi-variate Gaussian distribution. The prior distribution of $\mathbf{x}$ and $\boldsymbol{\theta}$ is contained within $p(\mathbf{x}, \boldsymbol{\theta})$. More detailed information about this method can be found in previous studies[54,58].

In the inversion we estimate the independent scaling factors for $\mathbf{x}$, and the hyperparameters $\boldsymbol{\theta}$ in each year. The prior distribution of $\mathbf{x}$, both for the scaling of the a priori emissions and boundary conditions, was assumed to follow a log-normal distribution with mean and standard deviation of 1, which is fairly uninformative, while reasonably constraining the scaling to realistic (one order of magnitude) emissions. The emissions were assumed constant during each year and estimated by adapting the scaling factor during the inversion. For boundary conditions, the magnitude of the TOMCAT mole fractions was scaled up and down in the inversion on each boundary (e.g. Lunt et al.[54]). The model-measurement uncertainties in the estimations consist of two parts, known measurement uncertainties, which is the repeatability of measurements, and the model error, which is unknown and needs to be estimated. The unknown model errors were estimated as hyperparameter $\boldsymbol{\theta}$ in the inversion, the prior distributions of which followed a uniform distribution and were estimated for each site, with their bounds set following a preliminary analysis. At SDZ, the model errors for flasks sampling and in situ measurement were estimated separately in the inversion.

To solve for the a posteriori parameters, a Markov Chain Monte-Carlo (MCMC) method was employed, following the approach of Say et al.[55]. Each output parameter and hyperparameter was sampled from Markov chains of $2.5 \times 10^5$ steps, which were constructed by a two-step sampler, using a No-U-Turn sampler (NUTS)[59] for the emissions and boundary conditions, and a slice sampler[60] for the hyperparameters. This system was implemented using PyMC3 software package[61]. The first 50,000 steps in the chain were removed as 'burn in' and $1.25 \times 10^5$ steps were used prior to sampling, and subsequently discarded, to tune the algorithm. We present our annual inversion results as the mean values and the corresponding 68% uncertainty intervals (16–84%, representing 1 s.d. for Gaussian uncertainties) derived from the posteriori distributions of emissions. The enhanced observations above baseline values can be reproduced well by the inferred emissions in this study (Supplementary Fig. 1).

Inversions with different magnitudes (0.5 to 2 times the initial a priori estimate) or spatial distributions (uniformly distributed or distributed by population) of a priori emissions were conducted to show that results are reasonably insensitive to the choice of the a priori emissions (Supplementary Fig. 2 and Supplementary Fig. 5). Emissions were also estimated without any a priori information except a positive constraint for the emissions for comparison (Supplementary Fig. 2b).

**Bottom-up estimation for $CH_2Cl_2$ emissions in China.** New bottom-up estimates were calculated following a previous study[17] with some modifications. Emissions of $CH_2Cl_2$ in China were assumed to originate from 5 sectors, (a) production leakage; (b) feedstock use; (c) pharmaceutical industry; (d) emissive solvent and (e) byproduct emissions. The emissions were calculated by Eq. (3),

$$E_i = A_i \times EF_i \qquad (3)$$

where $E_i$ is the emission of $CH_2Cl_2$ in sector i, $A_i$ is the activity level of $CH_2Cl_2$ in that sector, and $EF_i$ represents the corresponding emission factors. The emission factors and activity level data used in this study are provided in Supplementary Tables 3 and 4. The total production and consumption data were obtained from CCAIA[33]. For sector (a), the activity level is the total production of $CH_2Cl_2$ in each year. The activity level for sector (b) is the consumption of $CH_2Cl_2$ used as feedstock for HFC-32 production, derived from the HFC-32 production in each year. The activity level for the pharmaceutical industry, sector (c), is the consumption of $CH_2Cl_2$ in that sector, estimated by the overall consumption in each year, and the emission factor was estimated by the rate of solvent recovery and waste treatment in the pharmaceutical industry. All other end-uses were considered 100% emissive, which is called "emissive solvent", sector (d), in this study (the use as blowing agent for polyurethane foam was included in this sector). In these emissive sectors $CH_2Cl_2$ was assumed to be released completely within two years (50% in each year). Although for 2013, the emission from solvent sector was assumed to be equal to the total solvent consumption in this year due to the lack of data in the previous year. In this study, byproduct emission, sector (e), includes the emissions from coal production and combustion, and from iron and steel production. The byproduct emission sector makes only a minor contribution to the overall emissions[17,62]. Byproduct emissions were calculated by multiplying the activities in each process by the corresponding emission factors.

In the bottom-up inventory estimation, we assumed a normal distribution with 5% uncertainty for all the statistical activity data used in this study. A Monte-Carlo method with 100,000 samples was employed to calculate the bottom-up emissions and uncertainties.

**Global emissions estimation.** Global emissions of $CH_2Cl_2$ (2011–2019) were estimated using the AGAGE 12-box model[23,24], updating values previously published through 2016[1]. Briefly, the 12-box model divides the atmosphere into three vertical levels, bounded at the surface, 500 hPa and 200 hPa. Each level comprises of four latitude bands separated at 30° N, the equator and 30° S. The lifetime of $CH_2Cl_2$ in the 12-box model, determined primarily by reaction with an annually repeating OH field[23] and rate coefficients[63], was 6 months. Monthly baseline measurement data from five AGAGE background measurement sites[22], namely Mace Head, Ireland (53.3° N, 9.9° W), Trinidad Head, California, USA (41.1° N, 124.2° W), Ragged Point, Barbados (13.2° N, 59.4° W), Cape Matatula, American Samoa (14.2° S 170.7° W) and Cape Grim, Tasmania, Australia (40.7° S, 144.7° E), were simulated and emissions were inferred using a Bayesian method in which the emissions growth rate was constrained a priori[64]. This constraint was chosen to be very weak, such that the a priori year-to-year emissions change was assumed to be zero plus or minus 20% of the global emissions estimated by Xiao et al.[65]. Systematic uncertainties, including the error due to lifetime (20%), transport (1%) and calibration uncertainty (3%), are included in the emissions estimate[64] (although the difference between the AGAGE and NOAA calibration scales of ~10% is substantially larger than 3%, but the reasons for this difference are not known).

**Estimation for the impact of $CH_2Cl_2$ on Antarctic ozone recovery.** The impact of increasing $CH_2Cl_2$ on global ozone recovery has been investigated previously by Hossaini et al.[6]. In that study, three sensitivity scenarios were considered in order to explore a possible delay to Antarctic ozone hole recovery back to pre-1980s levels caused by $CH_2Cl_2$: in scenario 1, the future surface $CH_2Cl_2$ mole fractions would increase at the mean rate observed during 2004–2014, causing a 17–30 year delay in the Antarctic ozone hole return date; in scenario 2, an extreme growth scenario, the future surface $CH_2Cl_2$ mole fraction would continue to increase at the mean rate of 2012–2014, in which case the Antarctic ozone hole will not return to pre-1980s level by the end of this century; in scenario 3, the $CH_2Cl_2$ surface mole fractions would keep constant at the 2016 level, inducing a delay of ~5 years.

In our study, two scenarios are discussed based on two of the Hossaini et al.[6] scenarios. In the first scenario, future global $CH_2Cl_2$ mole fractions will continue to increase at the current mean rate observed in the study period (2011–2019), which is a "high growth" scenario. This scenario is approximately the same as a scenario where future global $CH_2Cl_2$ emissions continue to increase at the current mean rate observed in the study period (the future emissions will increase linearly at the current rate, if estimated by a one box model[66]). The mean growth rate of $CH_2Cl_2$ mole fractions in 30–90°N semi-hemisphere during 2011–2019 derived in this study (see Section Global emissions estimation above) is 2.63 (2.25–3.01) ppt yr$^{-1}$, similar to the growth rate in scenario 1 of Hossaini et al.[6], which is 2.85 ppt yr$^{-1}$ (30–60°N). This scenario indicates that we are currently following scenario 1 from Hossaini et al.[6], and if this continues in the future, a 17–30 year delay in Antarctic ozone hole recovery could occur.

In a second scenario, future global $CH_2Cl_2$ mole fractions will remain at 2019 values, which is a "moderate" scenario. Given the short lifetime of $CH_2Cl_2$, constant global mean mole fractions over timescales of around 1 year or longer implies constant emissions, to a very good approximation. The increase in both global emissions and emissions from China—the currently observed predominant source—seem to have slowed in recent years (see Results and Fig. 2a). Therefore, this "moderate" scenario would result from the plateau in emissions continuing into the future. Fortuitously for our calculation, the 2019 global mean mole fraction from the AGAGE network is very similar to the 2016 value from the NOAA network used to construct the scenarios in Hossaini et al.[6]. This is because calibration differences between the networks offset the growth during this period[1]. Therefore, we can assume that our "moderate" scenario is approximately the same as the scenario 3 in Hossaini et al.[6]. In this case, the estimated delay to Antarctic ozone hole recovery is ~5 years.

## Data availability

Measurement data of $CH_2Cl_2$ from AGAGE sites can be accessed at http://agage.mit.edu. Measurement data for the flask and in situ sites from CMA, and the inventory data used in the bottom-up analysis are provided in Supplementary information. Use of the CMA measurement data in publications, reports or presentations requires the users to contact B.Y. (yaob@cma.gov.cn) first to discuss your interests.

## Code availability

License to use NAME is available upon request to the UK met Office or upon request from A.J.M. (alistair.manning@metoffice.gov.uk). The code for the regional hierarchical Bayesian inversion, and all inputs and outputs are available upon request from M.A. (amdcese@pku.edu.cn) and M.R. (matt.rigby@bristol.ac.uk). The AGAGE 12-box model code is available upon request from M.R. (matt.rigby@bristol.ac.uk).

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

## Acknowledgements

This work was supported by the National Key Research and Development Program of China (Grant No. 2019YFC0214500). We are thankful for the stations personnel who have supported the in situ measurements and daily or weekly canister sampling at SDZ, WLG, LAN, LFS, XGL, JGJ, XGF, AKD, JSA. This work has benefited from the technical expertise of and assistance by the AGAGE (Advanced Global Atmospheric Gases Experiment) network including the Medusa GC/MS system technology, calibrations of $CH_2Cl_2$ measurements and network operation, as well as Dr. Martin Vollmer from Swiss Federal Laboratories for Materials Science and Technology. We acknowledge the support from members of Atmospheric Chemistry Research Group at University of Bristol. Measurements at the Mace Head, Trinidad Head, Ragged Point, Cape Matatula, and Cape Grim AGAGE stations are supported by the National Aeronautics and Space Administration (NASA) (grants NNX-16AC98G to MIT, and NNX16AC97G and NNX16AC96G to SIO). Support also comes from the UK Department for Business, Energy & Industrial Strategy (BEIS, Contract 1537/06/2018 to the University of Bristol) for Mace Head, the National Oceanic and Atmospheric Administration (NOAA, Contract RA-133-R15-CN-0008 and 1305M319CNRMJ0028 to the University of Bristol) for Ragged Point, and the Commonwealth Scientific and Industrial Research Organization (CSIRO) and the Bureau of Meteorology (Australia) for Cape Grim. R.H. is supported by a NERC Independent Research Fellowship (NE/N014375/1) and the NERC ISHOC project (NE/R004927/1). L.M.W. and M.R. received funding from NERC grants NE/M014851/1, NE/N016548/1 and NE/S004211/1, and L.M.W. received funding from the European Union's Horizon 2020 research and innovation programme under the Marie Skłodowska-Curie grant agreement No 101030750. A.L.G. was supported by U.K. Natural Environment Research Council (NERC) Independent Research Fellowship (NE/L010992/1).

## Author contributions

B.Y. and L.C. provided measurement data from the nine CMA sites. P.B.K., J.M., S.O'D., R.G.P., R.F.W. and D.Y. provided measurements from five AGAGE backgrounds sites. R.H. and T.C. provided a priori boundary condition values and contributed to the discussion on the impact to stratospheric ozone. M.A. conducted the inverse modelling and interpreted the results with the support of L.M.W., D.S., M.R. and A.L.G. M.R. provided the global emissions estimates from the AGAGE 12-box model. A.J.M. contributed to some of the NAME modeling. M.A. led the writing of the manuscript, with contributions from J.H., B.Y., M.R., L.M.W., D.S., J.M., R.G.P., R.F.W. and all other authors.

## Competing interests

The authors declare no competing interests.
