## [Peer Review File · Nature Communications]

REVIEWER COMMENTS

Reviewer #1 (Remarks to the Author):

This paper presents observations and analysis results based on dichloromethane (CH₂Cl₂) concentrations recorded in nine stations located around China. By using the measurements from stations, the authors provide the top-down time series of annual CH₂Cl₂ emissions from China and estimate global emissions growth. Most CH₂Cl₂ (~90%) in the atmosphere is emitted from anthropogenic activities. As an extended study of the bottom-up inventory analysis in China (Feng et al, AE, 2018), the basic analysis of CH₂Cl₂ is probably useful for understanding current situation of long-lived ozone-depleting substances, but there are several issues with the novelty on new findings (only the observed increased concentrations?) and environmental implication (How to control the potential sources in the future?) that need to be addressed before final publish. The sources for the estimated emissions growing from 232 Gg/year in 2011 to 627 Gg/year in China should be clarified. Since there are several major emission sources emitted CH₂Cl₂. However, China has employed stricter emission standards for most emission sources during this period. The authors should clarify the specific sources via combining with experimental evidences. Are the current known sources included in previous studied inventories from this paper and previous papers? Since some recent experimental papers also reported that some other emission sources also emitted CH₂Cl₂, such as the iron and steel plants. The authors should discuss the missing CH₂Cl₂ sources.

Similar as the previous comment. If the attribution from China had increased from ~30% 2011 to ~50% after 2012, the authors should detail the contribution of the emission sources.

Furthermore, the authors make the major conclusion from the nine station record data, but the global emissions should be estimated from measurements in worldwide stations via using the same estimation method. The authors should clarify this point.

Line 126-135: The discussion on the gap between the two bottom-up is very shallow. A detailed table for the new bottom-up inventory can be presented in the supporting information as for potential readers to check or follow, as well as the up-to-bottom inventory.

Line 149-159: The variation of the spatial emissions should be discussed with the local emissions with measurement evidence.

Reviewer #2 (Remarks to the Author):

This article presents new observational data from multiple sites in China of the VLSL CH₂Cl₂. These data are used to perform an inversion-based quantification of CH₂Cl₂ emissions in China for the recent period 2011-2019. The inversion-based estimate of emissions is compared favourably with a complementary "bottom up" estimate derived using new industry data, and the contribution of Chinese emissions to the growing global total is estimated. An evaluation of possible implications for ozone layer recovery, based on previously published scenarios, is also presented.

There has been much recent interest in the potential role of CH₂Cl₂ in the current and future evolution of the stratospheric ozone layer, and this work presents a significant advance (e.g. beyond the knowledge presented in the most recent Scientific Assessment of Ozone Depletion) in understanding of the temporal and spatial emissions in China, seemingly confirming this key contribution to global emissions.

The overall conclusions and claims are well supported by the modelling, data analysis and interpretation. The methodology appears sound and at the cutting edge of standards in the field. On the whole enough detail is provided in the methods for the work to be reproduced.

I found the article mostly clearly written, and the improvements suggested below (by line number) comprise only minor clarifications of wording.

Please note these line numbers refer to the pdf. There seem to be some differences in the Word version of the manuscript I had access to.

59 – “currently sparse” could be more precisely worded. Do they exist? Or the observational network is sparse?

119 – suggest writing “... emission in China ...” to avoid ambiguity

221 – it is not clear exactly where “up to a few years” comes from. I suggest, as already done in the Abstract, rephrasing this to better reflect the scenario-based assessment of a delay to ozone-layer recovery.

342 – “times” rather than “time”

395 – For clarity, suggest rewriting to that the “extreme growth scenario” text is in the same sentence as the text on scenario 2

523 – suggest “... China which occurred ...”

Reviewer #3 (Remarks to the Author):

The manuscript provides a detailed analysis of recent emissions of dichloromethane by China from measurements made in nine sites of the country and a top-down approach. Results are compared to a new bottom-up estimate of Chinese emissions, which although overestimated, is in better agreement with the top-down estimate. The major result of the study is a substantial increase (172%) of dichloromethane emissions by China over the 2011 – 2019 period, which is claimed to be the main source of global CH₂Cl₂ emission increase. As this very short-lived chlorine species is not regulated by the Montreal protocol, such an increase has potential impact on the recovery of the ozone layer. Another major finding is the identification of new CH₂Cl₂ emission regions in China linked to increased population and industrial activity. Methods used in the study are sound and based on proven methodologies for such measurements and analyses. The article is well written and well referenced. However, I have the following major issues with the manuscript that needs to be accounted for before eventual publication.

1. The global estimate of CH₂Cl₂ emission shown in Figure 1 is not sufficiently discussed in the context of other evaluations provided in e.g. the last WMO, 2018 Scientific Assessment of Ozone depletion or by the more recent study by Claxton et al. (2020). The latter, which is cited in the manuscript, estimates that combined Asian emission were equal to 1,045 Gg yr⁻¹ in 2017 (Table 3 of that article), while the present study estimates emissions by China alone of 627 Gg yr⁻¹ in 2019. It would be interesting to understand how the various estimates can be reconciled within the respective error bars.

2. The Claxton et al. analysis and global estimate of CH₂Cl₂ emissions uses a larger number of measurements, including those from flights campaigns, while the present uses measurements from global AGAGE sites only. The present manuscript lacks a discussion on the impact of the number of measurements used in the top down approach for the global estimate, which is important, considering the claim that Chinese emission increase accounts for the total global emission increase. The use of 6 months lifetime of CH₂Cl₂ used in the AGAGE box model for the global estimate as indicated in the Method section also warrants an additional discussion.

3. The authors claim that the “overall increase in CH₂Cl₂ emissions from China during the studied period explains the total global increase” (lines 99-104). Other authors indicate that emissions from the Indian subcontinent have increased substantially over the 2000s, e.g. Leedham-Elvidge, 2015 (<https://acp.copernicus.org/articles/15/1939/2015/>). A discussion of potential sources of

increase from other regions is thus needed before drawing such important conclusion about the contribution of China to the global increase of CH₂Cl₂ emissions.

4. A new bottom-up estimate of CH₂Cl₂ estimate is provided from newly obtained consumption and production data from the China Chlor-Alkali Industry Association (CCAIA). Since this estimate uses similar methods as in Feng et al. (2016), it is not clear why the new bottom up estimate so much differs from the previous one. Some assumptions linked to the emissivity of some sectorial consumption of CH₂Cl₂ need further discussion as they are derived from Feng et al (2016) without critical analysis.

5. In the Method section, there is a very detailed explanation of the methodology used to derive regional emissions but no real discussion of the sensitivity of each measurement site to the retrieved emission. Some results are given in supplementary figures 6-8 but without further analysis. This is needed as it can impact the final result on regional emissions from China.

6. Discussion of the impact of the dichloromethane increase on the Antarctic ozone hole completely relies on a sensitivity study by Hossainy et al., 2017 and is thus not a finding of this study. Considerations on possible future scenarios are based on new legislation in China that needs to be enforced, and on the feedstock use of CH₂Cl₂ for future production of HFC-32, which is rather hypothetical.

7. The organization of the manuscript was sometimes difficult to follow as some important information for understanding the results were given in the Method section that itself uses information from the supplementary section. A better distribution of major information between the main part of the article and the Method section would help better read the manuscript.

Response to reviewers' comments

Original manuscript number: NCOMMS-21-22567

Title: “Rapid increase in dichloromethane emissions from China inferred through atmospheric observations”

The text in italics is the reviewer's comment, followed by our response. During the revision of the manuscript following the reviewers' comments, we have redone some of the top-down inversions, and the bottom-up estimation. Some reported quantities may have changed slightly but there has been no change to the conclusions drawn.

Reviewer #1:

1. As an extended study of the bottom-up inventory analysis in China (Feng et al, AE, 2018), the basic analysis of CH₂Cl₂ is probably useful for understanding current situation of long-lived ozone-depleting substances, but there are several issues with the novelty on new findings (only the observed increased concentrations?) and environmental implication (How to control the potential sources in the future?) that need to be addressed before final publish.

Response: The core finding of this manuscript is the substantial increase of CH₂Cl₂ emissions from China, which could explain the overall global increase during the same period. This result is derived from inverse analysis of the observed concentration data in China. As stated in the introduction (line 64) this is the first such “top-down” time series of CH₂Cl₂ emissions from China.

In terms of the implications of our work for the control of CH₂Cl₂ in future, we show the potential environmental impact from CH₂Cl₂ by examining our findings in relation to two scenarios (see the first paragraph of “Discussion”). This analysis shows that CH₂Cl₂ is important for global ozone depletion and will likely become more important in the future. We also show that anthropogenic emissive solvent use is likely the main source for CH₂Cl₂ emissions in China, and that emissions from this sector have likely grown substantially (see “Results” lines 140-143, and the second paragraph of “Discussion”). This information is valuable for stakeholders interested in controlling future emissions. We discuss the current government limits on CH₂Cl₂ concentrations

in consumer products. We also note that there is currently no limitation on the production/consumption of CH₂Cl₂ and we discuss the expected future changes to CH₂Cl₂ demand (see the third paragraph of “Discussion”). We feel that going further than this and providing advice for future control is beyond the scope of this study.

2. The sources for the estimated emissions growing from 232 Gg/year in 2011 to 627 Gg/year in China should be clarified. Since there are several major emission sources emitted CH₂Cl₂. However, China has employed stricter emission standards for most emission sources during this period. The authors should clarify the specific sources via combining with experimental evidences.

Response: It is important to note that inverse modeling of atmospheric observations (i.e. experimental evidence) provides, primarily, an estimate of the net flux to the atmosphere from a wide area of the surface (~100s of km in scale). It is challenging to infer a sector-level breakdown of sources from atmospheric data, unless there are very distinct spatial or temporal patterns for these sources. Therefore, we have tried to infer the likely drivers of the emissions increase by examining: a) the bottom-up inventory data and, b) the spatial distributions of the top-down emissions. In the revised manuscript, we have expanded the discussion of the sources of CH₂Cl₂.

We have added the relevant discussion to the “Results” section, lines 140-143:

“According to the bottom-up inventory, the solvent sector accounts for more than 90% of overall CH₂Cl₂ emissions from China in all years and more than 85% of the overall increase during the study period, which indicates that the solvent sector is the dominant source for the emissions of CH₂Cl₂ from China.”

“Discussion” lines 198-202:

“The main regions of CH₂Cl₂ emissions in China and their increase over the study period are highly consistent with the locations of economically developed and industrialized regions, confirming a substantial anthropogenic source for emissions in China. This finding is supported by the bottom-up inventory in this study, where emissive solvent use of CH₂Cl₂ such as painting or adhesive use, accounts for most of the emissions and their increase.”

And lines 210-213:

“The increase in general chloromethanes production, which could be driven by the growing economy in China, would therefore be expected to cause an increase in CH₂Cl₂ emissions. This explains the strong correlation between chloromethanes emissions, Gross Domestic Product and the expanding chloromethanes production in China (Supplementary Fig. 6).”

A comprehensive discussion about the potential sources can be seen in the second paragraph of “Discussion” in the manuscript.

As the reviewer notes, there is now more strict regulation on CH₂Cl₂ in China. However, the regulation limits CH₂Cl₂ concentrations in the atmosphere or in consumer products, not overall production or consumption. Therefore, there is little regulation that would restrict the overall growth of CH₂Cl₂ production and emissions in China, provided this occurred without breaching concentration thresholds. In this case, an overall increase in demand may contribute to an overall emission increase.

3. Are the current known sources included in previous studied inventories from this paper and previous papers? Since some recent experimental papers also reported that some other emission sources also emitted CH₂Cl₂, such as the iron and steel plants. The authors should discuss the missing CH₂Cl₂ sources.

Response: Thank you for identifying a missing source. We have now added the emission of CH₂Cl₂ from iron and steel plants based on data from a recent paper¹ and the Chinese yearbook to the text (see “Methods” lines 397-400 and Supplementary Tables 3 and 4). This source is shown to be very small and does not make a substantial difference to the inventory results and the conclusions we draw:

Lines 397-400: “In this study, byproduct emission, sector (e), includes the emissions from coal production and combustion, and from iron and steel production. The byproduct emission sector makes only a minor contribution to the overall emissions^{17,62}. Byproduct emissions were calculated by multiplying the activities in each process by the corresponding emission factors.”

To the best of our knowledge, all of the primary sources have been included in our inventory estimate. Our new bottom-up inventory in this study used more comprehensive production/consumption data than previous estimates and is relatively

consistent with our top-down results (lines 143-152). This implies that the sources of CH₂Cl₂ should mainly come from the known sources included in the inventory. As mentioned in comment 2, increased anthropogenic use of CH₂Cl₂ as an emissive solvent is likely the predominant driver of the increase in emissions from China.

4. Similar as the previous comment. If the attribution from China had increased from ~30% 2011 to ~50% after 2012, the authors should detail the contribution of the emission sources.

Response: We thank the reviewer for the comment. Please see the explanation about how we inferred the sources for emissions increase in the response to comment 2.

5. Furthermore, the authors make the major conclusion from the nine station record data, but the global emissions should be estimated from measurements in worldwide stations via using the same estimation method. The authors should clarify this point.

Response: Both the regional and global emissions estimation frameworks we use are well-established and have been employed extensively in previously published studies²⁻⁴. We note that the two approaches are necessarily somewhat different. Our global emissions estimates are inferred from long-term trends in “background” concentrations (i.e., concentrations representative of the remote atmosphere, far from pollution sources) from a worldwide network (see section ‘Global emissions estimation’). In contrast, our regional approach infers emissions by comparing above-background pollution events from the national network to those simulated by a three-dimensional regional model. One previous study has used a global 3D model to infer global emissions, and their results are consistent with our global estimates, which uses a 2D model (lines 84-90). Given that our focus is on China, we opted to employ a high-resolution regional modelling framework here, rather than global one, because the regional approach allows higher resolution (and therefore, likely more accurate) simulation of atmospheric transport within China.

Lines 84-90:

“[Our] estimate shows a similar increase to recently published global emissions, derived using TOMCAT (a global 3D model) with measurement data from both the US National Oceanic and

Atmospheric Administration (NOAA) and AGAGE (adjusted to the NOAA calibration scale)²¹, or derived using a 12-box model with measurement data from NOAA alone¹. The different global emissions estimates agree within 1 s.d. uncertainty range, although the means differ by ~10-20 %, partly due to the differences in calibration scales between NOAA and AGAGE (~10% in CH₂Cl₂ measurement) and differences in the locations of the measurement sites used in the inversion¹.”

6. Line 126-135: The discussion on the gap between the two bottom-up is very shallow. A detailed table for the new bottom-up inventory can be presented in the supporting information as for potential readers to check or follow, as well as the up-to-bottom inventory.

Response: We have rearranged the discussion about the two bottom-up inventories (lines 130-143) to clarify the potential source of the gap:

“To reconcile the discrepancy with the bottom-up emission time series by Feng et al.¹⁷, a new bottom-up emission inventory was estimated following the method in Feng et al.¹⁷ (with some modifications, see Methods), using newly obtained production and consumption data for 2013-2019 from the China Chlor-Alkali Industry Association (CCAIA)³³. CCAIA is the only alkali industry association in China, and all chloromethanes companies/manufacturers are its members. These companies have the responsibility to share their production data with the association, ensuring that the dataset is representative of China’s entire chloro-alkali industry. These newly available production and consumption estimates are approximately twice as large as those used by Feng et al.¹⁷ (Supplementary Fig. 4a). Therefore, our new bottom-up emissions estimates are significantly larger (results shown in Fig. 2b, sectoral results in Supplementary Fig. 4b). The discrepancies between the two bottom-up inventories mainly originate from the emissive solvent sector, due to the difference in production and consumption data used in the two studies. According to the bottom-up inventory, the solvent sector accounts for more than 90% of overall CH₂Cl₂ emissions from China in all years and more than 85% of the overall increase during the study period, which indicates that the solvent sector is the dominant source for the emissions of CH₂Cl₂ from China.”

We note that the production and consumption data from Feng et al. (2016) were bought from a third-party advisory agent. Therefore, our production and consumption data, which originates from the CCAIA, should be more comprehensive.

We have revised the bottom-up estimate and re-organized the description of the

methodology to further clarify our approach (see method “Bottom-up estimation for CH₂Cl₂ emissions in China”). We have listed all the information used in the bottom-up inventory in the Supplementary Table 3-4. Detailed methodology for the top-down system have been provided in the “Methods” part and previous papers^{5,6}. In addition, we will provide all the code used in the emissions estimation.

7. Line 149-159: The variation of the spatial emissions should be discussed with the local emissions with measurement evidence.

Response: We are not sure what the reviewer means by “discussed with the local emissions with measurement evidence”. To clarify, the (well-established) regional inverse modelling approach is based on the interpretation of CH₂Cl₂ *concentration* data. To infer *emissions* from these data, a model of atmospheric transport is required (the NAME model in this case). The spatial distribution in emissions is derived in the inversion, by providing a “best fit” between the modelled and observed concentrations. We discuss the observations in the Supplementary Fig. 1, but the measured concentrations cannot be used as evidence for local emissions themselves.

For greenhouse gases and ozone depleting substances, in general, there are no independent local “measurements” of emissions that we can compare the inverse estimates to on comparable spatial scales.

Reviewer #2:

1. 59 – “currently sparse” could be more precisely worded. Do they exist? Or the observational network is sparse?

Response: It means that there are few current studies on top-down emission estimations within Asia. To avoid ambiguity, that line has been reworded to:

“However, there are few atmospheric observation-derived (top-down) estimates of CH₂Cl₂ emissions within Asia, and no estimates that span multiple years.”

2. 119 – suggest writing “... emission in China ...” to avoid ambiguity

Response: We thank the reviewer for the helpful suggestion. That line has been revised accordingly.

3. 221 – *it is not clear exactly where “up to a few years” comes from. I suggest, as already done in the Abstract, rephrasing this to better reflect the scenario-based assessment of a delay to ozone-layer recovery.*

Response: We thank the reviewer for the helpful suggestion. We have reworded the relevant sentences to:

“...which in this study is estimated to delay Antarctic ozone recovery by 5-30 years depending on different future scenarios...”

4. 342 – *“times” rather than “time”*

Response: We thank the reviewer for the helpful suggestion! The word in the manuscript has been revised.

5. 395 – *For clarity, suggest rewriting to that the “extreme growth scenario” text is in the same sentence as the text on scenario 2.*

Response: We thank the reviewer for the helpful suggestion! We have rearranged the lines to:

“...in scenario 1, the future surface CH₂Cl₂ mole fractions would increase at the mean rate observed during 2004-2014, causing a 17-30 year delay in the Antarctic ozone hole return date; in scenario 2, an extreme growth scenario, the future surface CH₂Cl₂ mole fraction would continue to increase at the mean rate of 2012-2014, in which case the Antarctic ozone hole will not return to pre-1980s level by the end of this century; in scenario 3...”

6. 523 – *suggest “... China which occurred ...”*

Response: We thank the reviewer for the helpful suggestion. The line in the manuscript has been revised.

Reviewer #3:

1. *The global estimate of CH₂Cl₂ emission shown in Figure 1 is not sufficiently discussed in the context of other evaluations provided in e.g. the last WMO, 2018 Scientific Assessment of Ozone depletion or by the more recent study by Claxton et al. (2020). The latter, which is cited in the manuscript, estimates that combined Asian emission were equal to 1,045 Gg yr⁻¹ in 2017 (Table 3 of that article), while the present study estimates emissions by China alone of 627 Gg yr⁻¹ in 2019. It would be interesting to understand how the various estimates can be reconciled within the respective error bars.*

Response: We thank the reviewer for the insightful comment. We have added the relevant discussion about our global emissions estimation compared with other evaluations at lines 84-90:

“[Our] estimate shows a similar increase to recently published global emissions, derived using TOMCAT (a global 3D model) with measurement data from both the US National Oceanic and Atmospheric Administration (NOAA) and AGAGE (adjusted to the NOAA calibration scale)²¹, or derived using a 12-box model with measurement data from NOAA alone¹. The different global emissions estimates agree within 1 s.d. uncertainty range, although the means differ by ~10-20 %, partly due to the differences in calibration scales between NOAA and AGAGE (~10% in CH₂Cl₂ measurement) and differences in the locations of the measurement sites used in the inversion¹.”

We discuss emissions from China in the context of emissions for Asia derived from Claxton et al. in lines 239-243:

“Using our estimated emissions for China and the estimates for India in 2016¹⁸, they together accounted for ~60% of the total Asian emission estimated by Claxton et al.²¹. The remaining ~40% likely originates from a combination of emissions from both land and ocean, although the exact emission breakdown is very uncertain due to the differences in methodology and measurement calibration used in the different studies.”

2. *The Claxton et al. analysis and global estimate of CH₂Cl₂ emissions uses a larger number of measurements, including those from flights campaigns, while the present uses measurements from global AGAGE sites only. The present manuscript lacks a discussion on the impact of the number of measurements used in the top-down approach for the global estimate, which is important, considering the claim that Chinese emission*

increase accounts for the total global emission increase. The use of 6 months lifetime of CH₂Cl₂ used in the AGAGE box model for the global estimate as indicated in the Method section also warrants an additional discussion.

Response: We have improved our discussion of the differences between our global results and the results from Claxton et al. (see the response to comment 1). Both the results follow similar trends and agree well within 1 s.d. uncertainty. If we take the results from Claxton et al. 2020 as reference, the overall increase of our derived emissions from China during 2011-2017 (the overlapping time period), 326 Gg yr⁻¹, is also of similar magnitude to the increase in Claxton's global estimate in the same time period, 285 Gg yr⁻¹.

There is ~10% calibration difference between NOAA and AGAGE measurement for CH₂Cl₂, which is a significant factor in the discrepancy observed between our work and Claxton et al.. To clarify, the measurements used in the inversion of Claxton et al. includes the NOAA and AGAGE observations, but no flight campaigns (the aircraft measurements were only used to test the results), and they recalculated all of the AGAGE observations based on the NOAA calibration scale (resulting in a ~10% increase in reported mole fraction). We believe that 5 AGAGE sites are sufficient to quantify global emissions of CH₂Cl₂ because: (1) estimating the global emissions with the 12-box model from these five global AGAGE sites is well-established, providing traceability to previous studies (e.g., Engel and Rigby, 2018⁴); (2) the five AGAGE stations are the primary sites where long-term baseline observations are available on the same calibration scale as the data from within China (a small number of additional baseline AGAGE stations are now available, but as yet, an assessment of the impact of including these stations on the global inversions has not been carried out); (3) The scope of our global inversion is much more limited than Claxton et al.: we estimated annual emissions from four surface boxes, instead of thousands of surface grid cells. Therefore, the use of a smaller number of measurements is justified in our case, and indeed, the inclusion of more measurements in our global emissions calculation does not reduce the primary sources of uncertainty, which are related to the systematic factors, lifetime and calibration scale.

We have added a statement about the lifetime and uncertainty in the 12-box model at

“Methods” part lines 418-421:

“Systematic uncertainties, including the error due to lifetime (20%), transport (1%) and calibration uncertainty (3%), are included in the emissions estimate⁶⁴ (although the difference between the AGAGE and NOAA calibration scales of ~10% is substantially larger than 3%, but the reasons for this difference are not known).”

The 6-month lifetime for CH₂Cl₂ is a consequence of the magnitude of the global OH field in the model (derived from a methyl chloroform inversion) and the OH-CH₂Cl₂ rate constant, which is taken from the JPL Assessment (see Methods). This value agrees well with the lifetime suggested in the WMO Scientific Assessment of Ozone Depletion⁷ (180 days). We include an estimate of the uncertainty due to this lifetime (where we assume 1 s.d. of 20%) in the global emissions uncertainty.

3. The authors claim that the “overall increase in CH₂Cl₂ emissions from China during the studied period explains the total global increase” (lines 99-104). Other authors indicate that emissions from the Indian subcontinent have increased substantially over the 2000s, e.g. Leedham-Elvidge, 2015 (<https://acp.copernicus.org/articles/15/1939/2015/>). A discussion of potential sources of increase from other regions is thus needed before drawing such important conclusion about the contribution of China to the global increase of CH₂Cl₂ emissions.

Response: We have changed the wording “explains the total” to “has the same magnitude” to avoid ambiguity. Furthermore, we added a discussion about the potential increase from India at lines 232-239:

“With emissions of CH₂Cl₂ in Europe and North America declining²¹, emissions from the developing world will have a growing impact on global CH₂Cl₂ emissions. In addition to the growth in emissions from China, a potential increase in CH₂Cl₂ emissions from India has been identified, based on emission estimates of 20.3 Gg yr⁻¹ in 2008⁴⁷ and 96.5 Gg yr⁻¹ in 2016¹⁸. The magnitude of this rise is relatively uncertain due to the methodological differences in the two studies, and it is small compared to the inferred increase from China. However, given that our study indicates that the growth in emissions from China is consistent with the coincident global rise, it is possible that any growth in India’s emissions has offset a decline from North America and Europe.”

4. A new bottom-up estimate of CH₂Cl₂ estimate is provided from newly obtained

consumption and production data from the China Chlor-Alkali Industry Association (CCAIA). Since this estimate uses similar methods as in Feng et al. (2018), it is not clear why the new bottom up estimate so much differs from the previous one. Some assumptions linked to the emissivity of some sectorial consumption of CH₂Cl₂ need further discussion as they are derived from Feng et al (2016) without critical analysis.

Response: We have rearranged and improved the discussion about the discrepancy between the two bottom-up inventories at lines 130-143.

“To reconcile the discrepancy with the bottom-up emission time series by Feng et al.¹⁷, a new bottom-up emission inventory was estimated following the method in Feng et al.¹⁷ (with some modifications, see Methods), using newly obtained production and consumption data for 2013-2019 from the China Chlor-Alkali Industry Association (CCAIA)³³. CCAIA is the only alkali industry association in China, and all chloromethanes companies/manufacturers are its members. These companies have the responsibility to share their production data with the association, ensuring that the dataset is representative of China’s entire chloro-alkali industry. These newly available production and consumption estimates are approximately twice as large as those used by Feng et al.¹⁷ (Supplementary Fig. 4a). Therefore, our new bottom-up emissions estimates are significantly larger (results shown in Fig. 2b, sectoral results in Supplementary Fig. 4b). The discrepancies between the two bottom-up inventories mainly originate from the emissive solvent sector, due to the difference in production and consumption data used in the two studies. According to the bottom-up inventory, the solvent sector accounts for more than 90% of overall CH₂Cl₂ emissions from China in all years and more than 85% of the overall increase during the study period, which indicates that the solvent sector is the dominant source for the emissions of CH₂Cl₂ from China.”

We have contacted the authors of Feng et al. (2018) to enquire about their production data, which were obtained from a survey of the chlor-alkali industry from a third-party private advisory institution, while our production and consumption data were obtained from China Chlor-Alkali Industry Association (CCAIA). We therefore assume that our production and consumption data are more reliable.

We have revised the methodology used in our bottom-up inventory with greater transparency. The revised approach is described in the revised “Methods” section (lines 380-404) and all the information used is listed in Supplementary Tables 3 and 4. Briefly, the emission factor in CH₂Cl₂ production leakage was changed to 0.5% as we think the

production process is a closed system, and this value is also recommended by the IPCC guidelines. The conversion efficiency of CH_2Cl_2 to HFC-32 has been changed according to expert advice, though the new value is quite similar to the previous value used. The emissions from the pharmaceutical industry, and from coal production and combustion were also recalculated. Emissions from steel and iron industry were added (see response to comment 3 of Reviewer #1). The uncertainty in the Monte Carlo sampling for the activity data has been changed to a Normal distribution. The updated inventory results were ~4-5% lower after these revisions.

5. In the Method section, there is a very detailed explanation of the methodology used to derive regional emissions but no real discussion of the sensitivity of each measurement site to the retrieved emission. Some results are given in supplementary figures 6-8 but without further analysis. This is needed as it can impact the final result on regional emissions from China.

Response: We have added a complementary inversion and strengthened the relevant discussion in the main text lines 103-106:

“Our modelled emissions are found to be relatively insensitive to the a priori emissions estimate and its uncertainty used in the inversion framework (Supplementary Fig. 2), and to the increasing number of measurement sites in the inversion throughout the study period (Supplementary Fig. 3).”

and “Methods” lines 287-293:

“The measurements made at these sites are sensitive to surface emissions from most of China (Supplementary Fig. 7), which makes it feasible to derive total national emissions from China using inverse modelling. The mean sensitivity from the sites in each year did not change substantially throughout the period, even though several new measurement sites were established and incorporated after 2017. The regional inversion was repeated using only observations from the five sites that were operational throughout the study period to show that the derived regional emissions are relatively insensitive to the number of measurement sites in the inversion (Supplementary Fig. 3).”

and the corresponding supporting results in Supplementary Fig. 3.

The results from this complementary sensitivity test are very similar to the results

presented in the main text. The overall trend is similar and the related conclusion drawn is not changed, though we found that the results of the latter three years are up to ~10% different. While more measurement sites in the latter years could lead to a more comprehensive sensitivity coverage, the measurements in the earlier years appear to be sufficient to capture the emissions from China relatively well (see Supplement Fig. 7, the sensitivity maps cover most of the key emission regions of China and don't vary significantly during the time period).

6. Discussion of the impact of the dichloromethane increase on the Antarctic ozone hole completely relies on a sensitivity study by Hossaini et al., 2017 and is thus not a finding of this study. Considerations on possible future scenarios are based on new legislation in China that needs to be enforced, and on the feedstock use of CH₂Cl₂ for future production of HFC-32, which is rather hypothetical.

Response: We thank the reviewer for the comment. As you note, the major and novel result of the work is that we find a substantial increase in CH₂Cl₂ emissions from China over 2011-2019, which has the same magnitude as the global increase. As your comments imply, there are uncertainties as to the future trajectory of CH₂Cl₂ emissions (e.g., due to any new legislation coming into force, importance of emissive/non-emissive applications in the future etc.). As such, it is important that future impacts on ozone are considered through a sensitivity framework while understanding of the factors that may influence future CH₂Cl₂ growth is refined. Importantly, in this study we show that CH₂Cl₂ growth is still within two of the sensitivity scenarios of CH₂Cl₂ growth presented by Hossaini et al. (2017). We then provided two hypothetical future scenarios based on the conclusions from Hossaini et al. (2017) for reference to emphasize the potential impact of future CH₂Cl₂ on the Antarctic ozone hole. While we do not claim that this is a new result, we think including this discussion provides important context to our main results and note that, for this work, there is little benefit to repeat the sensitivity analysis modelling work of Hossaini et al. for the reasons noted above.

7. The organization of the manuscript was sometimes difficult to follow as some important information for understanding the results were given in the Method section that itself uses information from the supplementary section. A better distribution of

major information between the main part of the article and the Method section would help better read the manuscript.

Response: We thank the reviewer for the comment. Several changes have been made on the manuscript, including: 1) re-arranging the method part and corresponding supporting information for the bottom-up inventory; 2) re-structuring the results part for the new bottom-up estimation, lines 130-143; 3) revising several pieces of text in the parentheses to improve the readability (e.g. lines 105 and 172); 4) re-structuring the method section on site information; 5) other minor adjustments on the structure of the sentences, marked as highlights. We feel these changes have made the paper clearer.

References

1. Ding, X. *et al.* Gaseous and particulate chlorine emissions from typical iron and steel industry in China. *J. Geophys. Res. Atmos.* **125**, e2020JD032729 (2020).
2. Rigby, M. *et al.* Increase in CFC-11 emissions from eastern China based on atmospheric observations. *Nature* **569**, 546–550 (2019).
3. Park, S. *et al.* A decline in emissions of CFC-11 and related chemicals from eastern China. *Nature* **590**, 433–437 (2021).
4. Engel, A. & Rigby, M. Chapter 1: Update on ozone-depleting substances (ODSs) and other gases of interest to the Montreal Protocol. in *Scientific Assessment of Ozone Depletion: 2018* vol. 58 (World Meteorological Organization, 2018).
5. Ganesan, A. L. *et al.* Characterization of uncertainties in atmospheric trace gas inversions using hierarchical Bayesian methods. *Atmos. Chem. Phys.* **14**, 3855–3864 (2014).
6. Say, D. *et al.* Emissions and marine boundary layer concentrations of unregulated chlorocarbons measured at Cape Point, South Africa. *Environ. Sci. Technol.* **54**, 10514–10523 (2020).
7. Burkholder, J. B. Appendix A: Summary of abundances, lifetimes, ozone depletion potentials (ODPs), radiative efficiencies (REs), global warming potentials (GWPs), and global temperature change potentials (GTPs). in *Scientific Assessment of Ozone Depletion: 2018* vol. 58 (World Meteorological Organization, 2018).

REVIEWERS' COMMENTS

Reviewer #1 (Remarks to the Author):

My concernings have been well addressed. No further comment.

Reviewer #2 (Remarks to the Author):

I found my earlier review comments were addressed satisfactorily.

Reviewer #3 (Remarks to the Author):

The revised manuscript has adequately addressed the issues raised in my review. The response to reviewers is very detailed and my suggestions have been taken into account. I thus suggest publication of the manuscript in its present form.